# Applications of Machine Learning
# in Constraining Multi-Scalar Models

**Darius Jurčiukonis[1]⋆**

**1** Vilnius University, Institute of Theoretical Physics and Astronomy

⋆ darius.jurciukonis@tfai.vu.lt

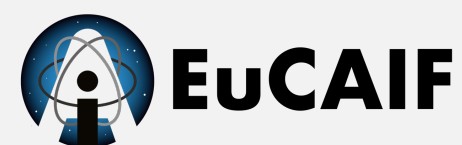

*The 2nd European AI for Fundamental Physics Conference (EuCAIFCon2025) Cagliari, Sardinia, 16-20 June 2025*

## Abstract

**Machine learning techniques are used to predict theoretical constraints such as unitarity and boundedness from below in extensions of the Standard Model. This approach has proven effective for models incorporating additional $SU(2)$ scalar multiplets, in particular the quadruplet and sixplet cases. High predictive performance is achieved through the use of suitable neural network architectures and well-prepared training datasets. Moreover, machine learning provides a substantial computational advantage, enabling significantly faster evaluations compared to scalar potential minimization.**

## 1 Introduction

Theoretical consistency in high-energy physics models is ensured by constraints such as unitarity (UNI) and bounded-from-below (BFB) conditions, which play a central role in establishing both physical viability and vacuum stability. Unitarity imposes restrictions on the behavior of scattering amplitudes, guaranteeing that they remain finite and well-defined. In contrast, the BFB condition ensures the stability of the vacuum by requiring that the scalar potential be bounded from below, thereby preventing the vacuum from decaying into states of arbitrarily lower energy.

Unitarity conditions can be evaluated analytically, through the computation of the eigenvalues of scattering matrices. Deriving analytical vacuum stability conditions is generally a challenging task and feasible only for relatively simple models. For more complex scenarios, one must rely on numerical minimization of the scalar potential, a process that is often computationally expensive.

In this study, two extensions of the Standard Model (SM) with additional $SU(2)$ multiplets, specifically the 4-plet and 6-plet cases, are investigated. The analytical UNI conditions derived in Ref. [1] together with the BFB conditions recently presented in Ref. [2] provide a framework to assess the reliability of machine learning (ML) approaches for this class of models. The

results are compared with those of the general two-Higgs-doublet model (G2HDM), for which the BFB conditions can be determined using a simple and precise algorithm [3, 4].

## 2  Strategy

Adopting the computational strategy proposed in Ref. [4], we train neural networks to predict the UNI and BFB constraints simultaneously. For that, we employ four fully connected neural networks, each consisting of eight layers as displayed in Fig. 1. The first network, net-1, comprises 128 neurons and is trained on raw data supplemented with an appropriate number of true samples. Net-1 is then used to generate refined training data for net-2, which has 256 neurons. Subsequently, net-3 and net-4 are trained on the same dataset as net-2 but with larger architectures of 512 and 1024 neurons, respectively. The outputs of net-3 and net-4 are further used to filter the predictions obtained from net-2. For this study, computations were performed on a desktop computer with the following configuration:

**System:**

- CPU: Intel® Core™ i9-13900K, 24 Cores (8P+16E) 2.2-5.8 GHz

- GPU: NVIDIA GeForce RTX 4090 (24 GB GDDR6X)

- RAM: G.Skill 128 GB 4 x 32 GB DDR5 6000 MHz

- SOFT: Ubuntu 22.04 LTS; Wolfram Mathematica 14.1

**ML parameters:**

- training data: $10^6$–$10^7$ samples

- training rounds: 500

- bach size: $2^{14} = 16\,384$

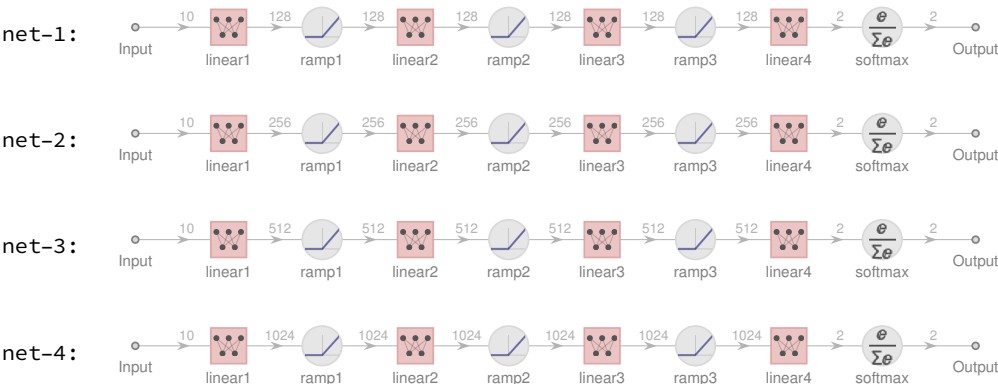

Figure 1: The architecture of the networks used in the analysis.

Network training was carried out on an GPU, while additional calculations were executed on an CPU. Since Wolfram Mathematica is widely used in the high-energy physics (HEP) community, all ML and numerical analyses were conducted within `Mathematica` to ensure compatibility with existing HEP packages. The functions and networks, together with computational examples are available at https://github.com/jurciukonis/ML-for-multiples.

## 3   Results

Randomly generated sets of potential parameters (raw data) contain only a small fraction of samples that satisfy the UNI conditions, the BFB conditions, or both, as shown in Table 1. Consequently, the raw data must be augmented with valid samples to ensure that the training dataset is sufficiently informative for effective network training.

| Model | UNI | BFB-I | BFB-II | UNI+BFB |
|-------|-----|-------|--------|---------|
| G2HDM | 1.02 | 23.3 | 3.25 | 0.033 |
| SM + 4plet | 12.8 | 30.0 | 17.3 | 2.2 |
| SM + 6plet | 5.4 | 22.2 | 6.07 | 0.33 |

Table 1: Percentage of true samples in the models with randomly generated parameters. The "BFB-I" column reports the fraction of true samples in the raw datasets, while the "BFB-II" column shows the corresponding fraction after refinement using UNI conditions. The last column reports the percentage of samples that simultaneously satisfy both UNI and BFB conditions.

Although the models under consideration admit analytical BFB conditions, they can also be evaluated through potential minimization, allowing direct comparison with computations performed using neural networks. The prediction of UNI and BFB conditions with ML can be accelerated by more than a factor of 400 in the case of the SM + 4plet model and by more than a factor of 600 in the case of the SM + 6plet model, relative to calculations based on potential minimization, as shown in Table 2. These predictions can subsequently be validated via the minimization procedure, while still maintaining computationally manageable times, as indicated in the last column of Table 2.

| Model | UNI+min. | neural nets | neural nets+min. | ratio-I | ratio-II |
|-------|----------|-------------|------------------|---------|----------|
| G2HDM | 131 | 1.7 | 35 | 77 | 3.7 |
| SM + 4plet | 53 | 0.12 | 19 | 441 | 2.7 |
| SM + 6plet | 496 | 0.76 | 45 | 653 | 11 |

Table 2: Computation times (in seconds) required to identify *1000 true samples*. The second column presents the computation times obtained using both UNI conditions and global minimization. The third column shows the corresponding times when employing neural networks alone, while the fourth column provides the times obtained by combining neural network predictions with subsequent verification through global minimization.

Since the raw data consist predominantly of false samples, even net-1 achieves very high accuracy in data classification. Therefore, to enable a more meaningful comparison, we focus on the fraction of true samples correctly classified by the networks, as verified using the analytical UNI + BFB conditions. For models with additional multiplets, the accuracy of net-1 remains relatively high; however, it can be substantially improved by employing more complex networks or network combinations, as shown in Table 3.

## 4   Conclusion

In this work, we investigated the reliability of machine learning methods for extensions of the Standard Model with additional $SU(2)$ multiplets. We demonstrated that ML techniques can

| Model | net-1 | net-2 | net-3,4 | net-2 – net-4 |
|-------|-------|-------|---------|---------------|
| G2HDM | 52-55 | 96-97 | 97-98 | > 99.0 |
| SM + 4plet | 98-99 | 98.8-99.1 | $\sim 99.2$ | > 99.5 |
| SM + 6plet | 92-94 | 98.8-99.2 | $\sim 99.0$ | > 99.5 |

Table 3: Percentage of true samples in the predicted results, validated against exact analytical UNI and BFB conditions.

efficiently predict UNI and BFB constraints in models with added 4-plet and 6-plet. This approach achieves high predictive accuracy while significantly reducing computation time compared to global minimization of the scalar potential. Although the models considered here are relatively simple, their parameter spaces become substantially more intricate when, for example, specific hypercharge assignments are introduced, as discussed in Refs. [5,6]. In such cases, neural networks could provide an effective tool for constraining the parameter space. Our results further indicate that this approach can be extended to more complex scenarios, such as the SM with additional 7-plet or 8-plet, for which analytical BFB conditions are not yet available.

## Acknowledgements

This work has received funding from the Research Council of Lithuania (LMTLT) under Contract No. S-CERN-24-2.

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
