# Peer review of "Applications of Machine Learning in Constraining Multi-Scalar Models"

_SciPost Physics Proceedings_

## Round 1 · Referee Report · Anonymous (Referee 1) · 2025-12-10

Disclosure of Generative AI use

The referee discloses that the following generative AI tools have been used in the preparation of this report:

By means of generative artifices (ChatGPT and Gemini), the reviewer’s rough notes were rendered into prose of more orderly disposition; yet the discernment, judgment, and scholarly responsibility rest wholly with the reviewer, upon whose mind alone these deliberations have been wrought.

Strengths

  1. Clear and motivated application of ML to a physics problem where substantial speedup is achievable.
  2. The availability of open-source code is very valuable for reproducibility.

Weaknesses

  1. The input to the neural networks is not described explicitly.
  2. No comment on the false positive rate, although this is relevant for a filtering-based workflow.
  3. Some aspects of the presentation could be improved for clarity (e.g., hardware list, Table 1).

Report

The manuscript presents a useful and clearly motivated application of neural networks to theoretical consistency conditions in multi-scalar models. The computational speedup is impressive, and the open-source implementation is appreciated. Some details are a bit brief in places, especially regarding the definition of the input features and the interpretation of the filtering performance. With these minor revisions, the contribution is suitable for publication in SciPost Physics Proceedings.

Requested changes

  1. Please specify which quantities form the 10-dimensional input to the networks.
  2. Add a short comment on the false positive rate to contextualize the filtering performance.
  3. Replace or shorten the bullet-point hardware list; a brief description plus indicative runtime or memory usage would be more informative.
  4. Consider adding a brief rationale for the chosen network architecture (depth, width).
  5. In Table 1, adding "%" symbols could improve readability; please also clarify why the fraction of true samples decreases from BFB-I to BFB-II.
  6. Correct the typo "bach size" to "batch size."
  7. The introduction cites exclusively the authors’ previous work; including one or two references to the broader HEP context would improve balance.

Recommendation

Ask for minor revision

---

## Editorial Decision

awaiting_resubmission